

# The superconducting clock-circuit:
# Improving the coherence of Josephson radiation
# beyond the thermodynamic uncertainty relation

**David Scheer$^\star$, Jonas Völler and Fabian Hassler**

Institute for Quantum Information, RWTH Aachen University, 52056 Aachen, Germany

$\star$ david.scheer@rwth-aachen.de

## Abstract

In the field of superconducting electronics, the on-chip generation of AC radiation is essential for further advancements. Although a Josephson junction can emit AC radiation from a purely DC voltage bias, the coherence of this radiation is significantly limited by Johnson-Nyquist noise. We relate this limitation to the thermodynamic uncertainty relation (TUR) in the field of stochastic thermodynamics. Recent findings indicate that the thermodynamic uncertainty relation can be broken by a classical pendulum clock. We demonstrate how the violation of the TUR can be used as a design principle for radiation sources by showing that a superconducting clock circuit emits coherent AC radiation from a DC bias.

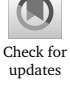

# 1  Introduction

In recent years, there has been growing interest in superconducting circuits with a high characteristic impedance above the quantum resistance $Z_0 \gtrsim R_Q = 2\pi\hbar/(2e)^2 = 6.45\,\mathrm{k\Omega}$, where $\hbar$ is the reduced Planck constant and $e$ is the electron charge. These devices have potential applications in quantum information processing, where they can be used in novel designs of robust qubits such as the fluxonium [1–3] and the 0-$\pi$ qubit [4–6] as well as in quantum metrology, where high impedance circuits are used to create Dual Shapiro steps [7–10]. While metrological precision is yet to be achieved, these steps are a promising candidate for completing the long-attempted redefinition of electrical units by connecting current to frequency [7]. In all of these applications, the supply of AC signals to the circuit is an essential requirement. However, it turns out that an external AC supply has to overcome major experimental challenges. The impedance mismatch between the device and the low-impedance biasing lines leads to a loss of power as well as a distortion of the driving pulses [11]. In addition, the biasing circuitry introduces stray capacitances that can destroy the required high-impedance environment altogether [12]. Furthermore, superconducting quantum processors require a large overhead of external AC lines, which poses challenges for scalability and miniaturization. A possible solution to these problems is the use of coherent on-chip radiation sources that only require an external DC bias [13]. Development in recent years brought forth a variety of on-chip microwave sources based on Josephson junctions that are coupled to a resonator. The implementations range from bright single photon sources [14, 15] to continuous wave devices which are based on either using a Josephson junction to pump a laser [16, 17] or the coherent self-synchronization of Josephson oscillations [18, 19].

The AC Josephson effect [20] is a natural contender for an AC source since the current-to-phase relation $I(Y) = I_c \sin(Y)$ of a Josephson junction produces sinusoidal current oscillations with an amplitude given by the critical current $I_c$ under a constant voltage bias $V = \hbar\dot{Y}/2e$. However, as a consequence of thermal fluctuations, a real voltage source—modeled by an ideal current bias $I_0$ in parallel with a large conductance $G$—produces Johnson-Nyquist noise that strongly limits the coherence of the resulting radiation [21–23]. It has been understood for a long time that the limited stability of the resulting radiation does not allow for the use of a bare junction as a stable on-chip source in practical applications. At fixed bias current $I_0$ and finite temperature $T$, the dephasing rate $\Gamma_\mathrm{J}$ of the Josephson oscillations fulfills [22, 23]

$$\Gamma_\mathrm{J} = \langle\!\langle Y^2(t) \rangle\!\rangle / t = \left(\frac{2e}{\hbar}\right)^2 2k_B T R_d^2 \frac{I_0}{V_0}, \tag{1}$$

with Boltzmann's constant $k_B$, the DC voltage drop $V_0$ across the junction, and the resulting differential resistance $R_d = dV_0/dI_0$. We denote the variance (mean) of $Y$ with respect to the thermal fluctuations by $\langle\langle Y^2 \rangle\rangle$ ($\langle Y \rangle$). For any bias current above the critical current $I_0 > I_c$, the differential resistance is larger than the Ohmic resistance of the biasing conductance $R_d G \geq 1$. As a result, $\Gamma_J$ is larger than the dephasing rate $\Gamma_0 = 8e^2 k_B T/(\hbar^2 G)$ for free diffusion across the bare Ohmic conductance. With the oscillation frequency $\omega_0 = 2eV_0/\hbar$, this results in an upper bound $Q_0 = \omega_0/\Gamma_0 = \hbar\omega_0 G R_Q/(4\pi k_B T)$ for the quality factor of Josephson oscillations.

This limitation can be viewed as a direct result of the Thermodynamic Uncertainty Relation (TUR) [24] from the field of stochastic thermodynamics. The TUR describes a trade-off between the precision of general integrated currents and the entropy production of a Markovian system in a non-equilibrium steady state. The statement of the TUR is given by

$$\frac{\langle\langle Y^2(t) \rangle\rangle}{\langle Y(t) \rangle^2} \sigma t \geq 2k_B \,, \tag{2}$$

with the entropy production rate $\sigma$ as a measure of the energetic cost required to maintain the corresponding non-equilibrium steady state. In the case of a current bias, the entropy production rate of a steady state is determined by the average power output of the source which is dissipated into the heat bath $\sigma = \hbar I_0 \langle Y(t) \rangle/(2eTt)$. This consideration is analogous to the one presented in [25].

While originally formulated for systems with discrete states [26], the TUR is also applicable to overdamped Langevin equations [27, 28] which are the common framework to describe Josephson radiation. While local violations of the TUR can be achieved through a phase locking to an external signal [29], as used to create Shapiro steps [30, 31], a self-sustained oscillation with high precision is not possible for overdamped systems. However, it was recently shown that the TUR does not hold for underdamped systems since it can be broken by a classical pendulum clock [25].

In this paper, we present a superconducting clock circuit that functions as a coherent self-sustained oscillator based on the AC Josephson effect. The minimal model of a pendulum clock presented in [25] consists of a counter and an oscillator, which are coupled by an escapement. The escapement is the crucial component of pendulum clocks since it provides an effective protection against environmental disturbance that established the pendulum clock as the precision standard in timekeeping for centuries [32]. We show that the circuit in Fig. 1 realizes a pendulum clock with a simple escapement potential. Starting from a quantum mechanical description of the oscillator, we perform the classical limit of large photon numbers to derive an effective model in terms of Adler-type equations. Similar to [33], we identify the resulting synchronization as the origin of the TUR violation of the counter. Finally, we identify a parameter that allows for increased coherence of the oscillator compared to a bare Josephson junction which allows for usage of the circuit as a stable on-chip radiation source.

## 2 Classical model

In a minimal setting, a pendulum clock requires two degrees of freedom—an underdamped oscillator $X$ that supplies a stable periodic motion and an overdamped counter $Y$ that moves at a constant velocity determined by the frequency of the oscillation. The crucial property of a clock is the fact that it produces stable oscillations at the frequency of the oscillator without relying on an external signal. This can be achieved through an escapement potential[1]

---

[1]In [25], the escapement potential is of the form $V_c(X, Y) = \kappa[X - \sin(Y)]^2/2$. This potential acts like a spring that enforces the motion of the clock along a trajectory $X(Y)$ that corresponds to a pendulum clock. However, the most important term for this is the coupling term corresponding to our escapement potential. The other terms mostly renormalize the frequency of the oscillator and the counting velocity which for the most part just complicates the analytical treatment of the problem as well as the construction of an appropriate circuit.

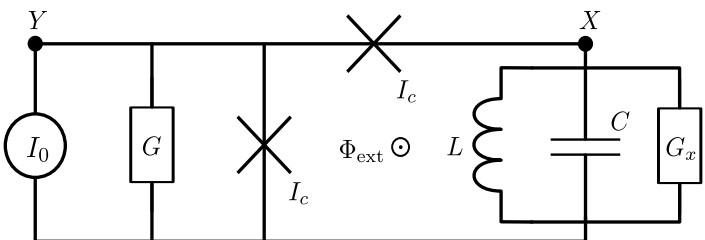

Figure 1: Superconducting clock circuit consisting of a Josephson junction with a real voltage bias that is coupled to an $RLC$-resonator via a second identical Josephson junction. At half a magnetic flux quantum threaded through the resulting loop, the circuit realizes an escapement coupling.

$V_c(X,Y) = -\kappa X \sin Y$ with a coupling parameter $\kappa$ that fulfills a twofold purpose [25]. On the one hand, it translates a linear motion of the counter to a periodic drive on the oscillator which is necessary to maintain a steady oscillation. On the other hand, it regulates the velocity of the counter through a constant force caused by a down-conversion of the oscillation.

A minimal model for a classical pendulum clock with a counter $Y$ and an oscillator $X$ is given by [25]

$$\Gamma \dot{Y} = f + \kappa X \cos Y\,, \qquad \ddot{X} + \gamma \dot{X} + \omega_0^2 X = \kappa \sin Y\,, \tag{3}$$

where the counter performs an overdamped motion with a damping constant $\Gamma$ and a constant force $f$ provided by an energy supply akin to the motion of a weight in a grandfather clock that is pulled down by gravity. The resulting terminal velocity enables to relate the position of the counter to the time that has passed. In the presence of thermal fluctuations arising from the dissipation, the constant force obtains a stochastic component that limits the accuracy of the counter according to the TUR. To allow for more precise timekeeping, the counter is coupled to an underdamped oscillator with frequency $\omega_0$ and a small damping constant $\gamma \ll \omega_0$ corresponding to a pendulum. The inertia of the oscillator $\ddot{X}$ allows the clock to achieve precision beyond the TUR since it only holds for overdamped dynamics.

The essential dynamics of the clock can be captured by the ansatz of a regulated linear motion $Y = \omega_0 t + \theta(t)$ for the counter and an oscillation with a time-dependent amplitude and phase for the oscillator $X = \text{Re}[A(t)e^{-i(\omega_0 t + \varphi(t))}]$. The external force can be adjusted to achieve a synchronization between the counter and the oscillator, where the phase variables become constant in time with $\dot{\theta} = \dot{\varphi} = 0$. The resulting counting velocity is determined by the frequency of the oscillator with $\dot{Y} = \omega_0$ leading to a resonant drive. This yields a stable oscillation with frequency $\omega_0$ and a steady state amplitude $\kappa/(\omega_0 \gamma)$ that does not require an external periodic drive.

## 3 Circuit setup

In Fig. 1, we present a superconducting circuit that implements the model of a pendulum clock. It consists of a real voltage source with a bias current $I_0$ and a biasing conductance $G$ that is coupled to a parallel $RLC$-resonator with inductance $L$, capacitance $C$, and conductance $G_x$ that models the photon loss from the resonator. The counting degree of freedom is given by the superconducting phase $Y$ that corresponds to the integrated voltage drop across the conductance with $V_0 = \hbar \dot{Y}/(2e)$. The oscillating degree of freedom is the superconducting phase $X$ at the resonator. The bare oscillations have a natural frequency $\omega_0 = \sqrt{LC}^{-1}$ with a typical amplitude given by the light-matter coupling constant $r = \pi Z_0/R_Q$ that compares the impedance of the resonator $Z_0 = \sqrt{L/C}$ to the quantum resistance.

We realize the escapement coupling through a pair of Josephson junctions with equal critical currents $I_c$, one in parallel with the biasing conductance and the other in series with the resonator. By threading an external magnetic flux $\Phi_{\text{ext}}$ through the resulting loop, we realize a tunable coupling potential

$$V_c = -\frac{\Phi_0 I_c}{2\pi}\left[\cos\left(Y - 2\pi\frac{\Phi_{\text{ext}}}{\Phi_0}\right) + \cos(Y - X)\right], \tag{4}$$

where $\Phi_0 = \frac{h}{2e}$ is the superconducting flux quantum. By tuning the external flux to half a flux quantum,[2] we realize a coupling potential that corresponds to the escapement potential in Eq. (3) for small oscillation amplitudes $X \ll 2\pi$. A similar circuit has been studied as an on-chip spectrometer [35, 36] that also exhibits a clock-like resonance within its absorption lines.

## 4 Quantum description

In the regime of an underdamped oscillator $\gamma = \omega_0 Z_0 G_x \ll \omega_0$, a description of the system needs to account for quantum effects. In App. A.1, we derive a Hamiltonian action for the circuit that allows for a quantum mechanical treatment of the system without the resistive elements. Since a Hamiltonian description cannot account for the dissipative dynamics of the circuit, we extend it to a Keldysh path integral in App. A.2.

We can obtain effective equations of motion for the circuit by making a rotating-wave ansatz for the oscillator $\hat{X} = \sqrt{r}(\hat{a}e^{-i\omega_0 t} + \hat{a}^\dagger e^{i\omega_0 t})$, where the variations of the photon annihilation operator of the resonator $\hat{a}$ are slow on the scale of $\omega_0$. In the rotating-wave approximation, the state $\hat{\rho}$ of the oscillator evolves according to a Lindblad equation

$$\dot{\hat{\rho}} = -\frac{i}{\hbar}[\hat{H}_{\text{RW}}, \hat{\rho}] + \gamma(n_0 + 1)\mathcal{J}[\hat{a}](\hat{\rho}) + \gamma n_0 \mathcal{J}[\hat{a}^\dagger](\hat{\rho}), \tag{5}$$

with photon emission and absorption processes described by jump operators $\mathcal{J}[\hat{O}](\hat{\rho}) = \hat{O}\hat{\rho}\hat{O}^\dagger - \frac{1}{2}\{\hat{O}^\dagger\hat{O}, \hat{\rho}\}$. In the case without coupling, the dissipation enforces a thermal equilibrium state with a Bose-Einstein occupation $n_0 = 1/\{\exp[\hbar\omega_0/(k_B T)] - 1\}$. We describe the escapement coupling by a rotating-wave Hamiltonian $\hat{H}_{\text{RW}}$ that depends on the state of the counter with a detailed derivation given in App. A.3.

For the description of the overdamped counter, we consider the regime of localized phase with $R_Q G \gg 1$ where the quantum fluctuations of the counter $\hat{Y}$ lie far below $2\pi$ [12, 37]. In this regime, the motion is well described by a Langevin equation for the expectation value $Y = \langle\hat{Y}\rangle$. Motivated by our classical intuition for a clock, we expect linear growth for the counter with a velocity regulated by the oscillator. We make the ansatz $Y(t) = \omega_0 t + \theta(t)$. In the regime of small junction capacitances with $C'/C \ll 1$ and $\omega_0 C' \ll G$, which we also discuss in App. A.3, the deviation $\theta(t)$ from the resonant counting motion follows an overdamped Langevin equation

$$\dot{\theta} = \Delta\omega - \frac{2\omega_0 r}{\hbar\Gamma}\text{Tr}\left(\frac{\partial\hat{H}_{\text{RW}}}{\partial\theta}\hat{\rho}\right) + \xi(t), \tag{6}$$

with a damping rate $\Gamma = \omega_0 Z_0 G$ and a stochastic force $\xi(t)$ that arises from the fluctuation dissipation theorem with $\langle\xi(t)\xi(t')\rangle = 4k_B T\omega_0 r/(\hbar\Gamma)\delta(t - t')$. Note that the density matrix $\hat{\rho}$ also depends on the trajectory of the stochastic force since the coupling Hamiltonian is conditioned on the state of the counter. The bias current corresponds to a constant force

---

[2]Note that while an external magnetic field can break the requirements for the TUR [34] this does not happen at $\Phi_{\text{ext}} \bmod \Phi_0 = 1/2$ since half a flux quantum preserves the time-reversal symmetry of the system.

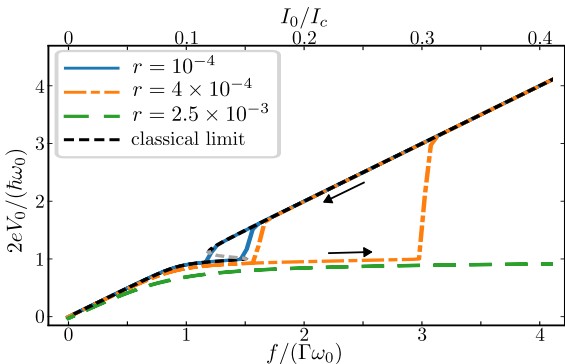

Figure 2: Resulting IV curves for varying light-matter coupling $r$ with effective parameters $\kappa/\omega_0^2 = 0.01$, $\gamma/\omega_0 = 0.1$, and $\Gamma/\omega_0 = 10^{-3}$. The curves show a pronounced voltage plateau at a voltage corresponding to $\omega_0$ with a hysteretic behavior in the plateau region. At $r = 10^{-4}$ (solid) the numerics match the analytical result for the classical limit (black dashed) up to the unstable solution in the hysteretic region (gray dashed).

$f = \omega_0 r I_0/e$ that enters into the frequency mismatch $\Delta\omega = f/\Gamma - \omega_0$. The coupling Hamiltonian realizes a conservative force that depends on the state of the oscillator. For large photon numbers $\langle n \rangle_\rho = \mathrm{Tr}(\hat{a}^\dagger \hat{a} \hat{\rho}) \gg 1$ and small light-matter coupling $r\langle n \rangle_\rho \ll 1$, we can approximate the Hamiltonian by

$$\frac{\hat{H}_{\mathrm{RW}}}{\hbar} = i\frac{\kappa}{4\omega_0\sqrt{r}}(\hat{a}e^{i\theta(t)} - \hat{a}^\dagger e^{-i\theta(t)}), \qquad (7)$$

with a coupling parameter $\kappa = \omega_0 r I_c/e$. Note that the expectation value with respect to the density matrix of the oscillator $\langle . \rangle_\rho$ is still conditioned on the thermal fluctuations $\xi$. The linear form of the Hamiltonian leads to closed equations of motion for the expectation values of the photon operators which we present in more detail in App. A.4. On resonance, the steady state photon number is given by $\langle n \rangle_\rho = n_0 + n_{\mathrm{coh}}$ with the coherent contribution

$$n_{\mathrm{coh}} = \frac{\kappa^2}{4r\omega_0^2\gamma^2}. \qquad (8)$$

To verify the validity of our approximation, we simulate the voltage drop of the counter, as a function of the external force $f$, solving Eq. (5) and Eq. (6) with the full rotating-rotating-wave Hamiltonian given by Eq.(A.11) in App. A.3 at zero temperature with varying light-matter coupling $r$. We compare the simulation to our analytical results for the classical limit with small $r$ and large photon numbers. To achieve a proper classical limit, we rescale the circuit parameters with $r$ to keep the effective parameters at a constant value of $\kappa/\omega_0^2 = 0.01$, $\gamma/\omega_0 = 0.1$, and $\Gamma/\omega_0 = 10^{-3}$. According to Eq. (8), the photon number increases with decreasing $r$. In Fig. 2, we show the resulting IV curves. For all values of $r$, the curves exhibit a voltage plateau corresponding to resonance with the oscillator, with hysteretic features in the plateau region indicating a competition between resonance with the oscillator and an Ohmic behavior of the counter. At $r = 10^{-4}$, the photon number is sufficiently large for the full model to reproduce the analytical results for the classical limit. For increasing $r$, the width of the synchronization plateau is increased. We attribute this to a stronger interaction between the photons in the resonator and the counter, which facilitates stronger synchronization effects.



## 5 Synchronization

At sufficiently small temperatures $rn_0 \ll \kappa^2/(\omega_0\gamma)^2$, we can describe the oscillator by a coherent state with a complex amplitude $2\sqrt{r}\,\mathrm{Tr}(\hat{a}\hat{\rho}) = Ae^{-i\varphi}$. In this state, the oscillating superconducting phase follows a trajectory $\mathrm{Tr}(\hat{X}\hat{\rho}) = \mathrm{Re}[Ae^{-i(\omega_0 t+\varphi)}]$. In this regime, the circuit is described by a set of Adler-type equations

$$\dot{\theta} = \Delta\omega + \frac{\kappa}{2\Gamma}A\cos(\varphi-\theta) + \xi(t), \qquad \dot{A} + i\dot{\varphi}A = -\frac{\gamma}{2}A - \frac{\kappa}{2\omega_0}e^{-i(\varphi-\theta)}, \tag{9}$$

which, up to the noise term $\xi(t)$, can also be derived from the purely classical model of a pendulum clock as we show in App. B.1. This shows that the circuit reproduces the model of a classical pendulum clock.

Eq. (9) illustrates that the key feature of the clock is a synchronization between the motion of the counter and the oscillator with a phase locking $\varphi(t) - \theta(t) = \varphi_0 - \theta_0$. Note that the individual phases $\varphi$ and $\theta$ do not need to be slow variables of the scale of $\omega_0$ as long as their difference is slow. This allows for substantial deviations from the bare frequency even within the rotating-wave approximation. The steady state solutions are given by

$$A_0 = -\frac{\kappa}{\gamma\omega_0\sqrt{1+\chi^2}}, \qquad \dot{\theta} = \dot{\varphi} = -\frac{\gamma}{2}\chi, \tag{10}$$

where $\chi = \tan(\varphi_0 - \theta_0)$ fulfills the synchronization condition

$$(\chi + 2\Delta\omega/\gamma)(\chi^2 + 1) = \frac{\kappa^2}{\omega_0\Gamma\gamma^2} = 2\beta. \tag{11}$$

The synchronization strength $\beta$ determines the range of bias currents, where the deviation from the resonance is small $\chi \ll 1$ and the oscillator enforces its frequency $\omega_0$ on the counter. This corresponds to a voltage plateau for the counter with $V_0 \approx \hbar\omega_0/(2e)$. For $\beta > 4/\sqrt{27}$, Eq. (11) allows for multivalued solutions that, similar to a Duffing oscillator, lead to a hysteretic behavior of the circuit. For the additional stable solution in the plateau region, the velocity of the counter remains unaffected by the oscillator which exhibits a small amplitude due to the resulting off-resonant drive. This behavior also occurs naturally outside the plateau region.

## 6 TUR violation

As the resonance plateau in the IV curve corresponds to a differential resistance much smaller than the biasing resistance, analogy with Eq. (1) suggests a strong TUR violation for the counter. Furthermore, we expect the oscillations of $X$ to be especially stable on resonance. We numerically analyze the behavior of the circuit at finite temperature $\epsilon/\omega_0 = 4k_B Tr/(\hbar\Gamma) = 5 \times 10^{-4}$ to show an explicit violation of the TUR for the counter and an increase in coherence for the oscillator. We consider the classical limit of large photon numbers with the same circuit parameters as in Fig. 2. In Fig. 3(a) and Fig. 3(b), we show the resulting IV curve along with the resulting differential resistance $R_d$ and the uncertainty product $\langle\!\langle Y^2(t)\rangle\!\rangle\sigma t/\langle Y(t)\rangle^2$ in comparison with an Ohmic resistance. As for the bare junction, the uncertainty product is proportional to the square of the differential resistance. As the lower bound for the variance given by the TUR corresponds to an Ohmic resistance, the plateau exhibits a violation of the TUR that gets stronger with a decreasing slope, while the TUR is recovered out of resonance. The circuit also exhibits a second small violation of the TUR that we attribute to a synchronization at half frequency. In the classical limit, we can obtain analytical insight into the circuit properties from Eq. (9), see App. B.2 for the detailed calculation.

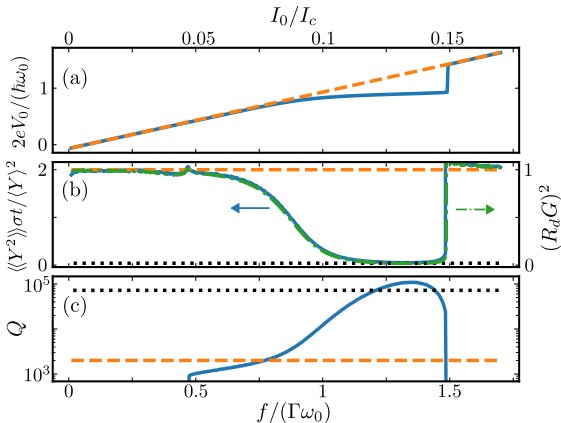

Figure 3: (a) IV curve (solid) in the classical limit at finite temperature $\epsilon/\omega_0 = 4k_B Tr/(\hbar\Gamma) = 5 \times 10^{-4}$. The circuit parameters are the same parameters as in Fig. 2 resulting in $\beta = 5$. On resonance, the curve deviates from an Ohmic characteristic (dashed). (b) The resulting uncertainty product in units of $k_B$ (solid) and the square of the differential resistance (dash-dotted) along the IV curve. Both curves coincide with one another, which shows that the uncertainty product is closely related to the differential resistance. Therefore, at differential resistances below the biasing resistance (dashed), the TUR (dashed) is violated with a new lower bound of $2k_B\gamma/(\beta\omega_0)$ (dotted). (c) Quality factor $Q$ of the emitted radiation (solid). Off resonance, $Q$ is of the order of $10^{-9}$ (not shown) and on resonance, it gets close to the maximum value $Q_0$ (dashed) given by the TUR which it exceeds by about two orders of magnitude roughly corresponding to a factor of $(1 + \beta)^2$ (dotted).

For $\beta \gg 1$, we obtain an estimate for the minimal long-time value of the uncertainty product given by

$$\frac{\langle\langle Y^2(t) \rangle\rangle}{\langle Y(t) \rangle^2} \sigma t \gtrsim 2k_B \frac{\gamma}{\beta\omega_0}, \tag{12}$$

that is determined by the synchronization strength $\beta$ and the quality factor of the resonator $\omega_0/\gamma$, which is in good accordance with the numerical uncertainty product shown in Fig. 3(b). Therefore, the counter produces a DC voltage with stability that is not limited by the TUR.

To assess the stability of the radiation emitted from the clock circuit, we consider the dephasing rate of the oscillator $\Gamma_C = \langle\langle \varphi^2(t) \rangle\rangle/t$. Together with the oscillation frequency $\omega = 2eV_0/\hbar$ along the IV curve, it yields a quality factor $Q_C = \omega/\Gamma_C$. In Fig 3(c), we show the numerical results for $Q_C$. Away from resonance, $Q_C$ is much smaller than the maximum quality factor $Q_0$ at frequency $\omega_0$ that is imposed by the TUR. However, at a resonant bias, the quality factor increases up to two orders of magnitude compared to $Q_0$. From Eq. (9), we obtain an estimate of the maximum quality factor given by

$$Q_C \lesssim Q_0(1 + \beta)^2, \tag{13}$$

which is in accordance with the numerical results in Fig. 3(c). This shows that the clock circuit can be used as an on-chip radiation source with a coherence beyond the limits of the TUR where the coherence increases with the square of the synchronization strength $\beta$.

# 7 Experimental parameters

In order to increase coherence, we have to increase the synchronization strength in a way that is consistent with the classical regime of large photon numbers and small light-matter coupling. Using $\beta = 2r n_{\text{coh}} \omega_0 / \Gamma$, we can obtain higher coherence in the classical regime by increasing the ratio $\omega_0/\Gamma$ without leaving the overdamped regime $\omega_0/\Gamma \ll r^{-1}$. We achieve an improvement of $1 \ll \beta \lesssim n_{\text{coh}}$ under the condition of $r^{-1} \gg n_{\text{coh}} \gg GR_Q \gg 1$. Since the diffusion constant $\epsilon$ also increases with $\omega_0/\Gamma$, the overall quality factor increases linearly in $\omega_0/\Gamma = 1/(Z_0 G)$. Since $G$ can not be made arbitrarily small in the overdamped regime, the vital parameter for coherence in the classical regime is the impedance $Z_0$ of the resonator.

At currently achievable impedances of $Z_0 \approx 5\,\Omega$,[3] the resulting light-matter coupling is too large to obtain coherence properties comparable to [18] in the fully classical regime with small non-linearities. Instead, the currently implementable devices can operate at the border of the validity of our approximations. As shown in Fig. 2, the voltage plateau corresponding to the synchronization physics and the TUR violation is robust even outside the classical regime. At increased light-matter coupling, the IV curves even exhibit a flatter, more pronounced voltage plateau. This indicates a potential use as an alteration of the single photon source presented in [14] where the thermal fluctuations of the bias voltage can be suppressed in order to increase coherence. Therefore, we expect the analytical results for the classical case to hold up reasonably well at their border of validity. In terms of the circuit parameters the overall maximum quality factor is given by

$$Q_C \approx Q_0 \beta^2 = \frac{\hbar \omega_0}{4 k_B T} \frac{G_Q^3}{G G_x^4 Z_0^2} \left( \frac{\pi I_c}{\omega_0 e} \right)^4, \tag{14}$$

with an output power

$$P \approx \hbar \omega_0 n_{\text{coh}} \gamma = \frac{I_c^2}{2 G_x}. \tag{15}$$

By increasing the load impedance $G_x^{-1}$, both the quality factor of the radiation and the output power are increased. This shows that our circuit is a promising candidate for an on-chip source in high-impedance experiments. While the results also suggest that an improvement of the critical current would be very beneficial for the operation of the circuit, they rely on a controlled photon number which increases too strongly with the critical current. We consider a set of feasible parameters at the borderline of $r n_{\text{coh}} = 1$ with $I_c = 0.2\,\mu\text{A}$, $G^{-1} = 1\,\text{k}\Omega$, $G_x^{-1} = 50\,\Omega$ and a resonator with $\omega_0 = 2\pi \times 5\,\text{GHz}$ and $Z_0 = 5\,\Omega$ at a temperature $T = 10\,\text{mK}$. Note that we use a load impedance of $50\,\Omega$ for better comparison with sources used in a typical RF setting. For these parameters, we obtain a linewidth of $\delta\omega = \omega_0/Q_C = 2\pi \times 4\,\text{kHz}$ at an output power of $1\,\text{pW}$. While the linewidth is the same as for the source in [18], the output power is reduced by one order of magnitude. We expect however, that further improvements could arise from an increased critical current which is at a value of $I_c = 10\,\mu\text{A}$ in [18].

In our analysis of the escapement potential, we have relied on equal critical currents $I_c$ for both Josephson junctions. While this is not exactly fulfilled in an experimental setting, the clock-like resonances in the experiments of [35] indicate that the escapement should remain functional within experimental accuracy. In App. C, we show that the circuit retains a smaller clock-like resonance even at a mismatch of 10% between the critical currents. If necessary, one junction can also be replaced by a SQUID to enforce symmetric critical currents.

---

[3]S. Lotkhov and F. Kaap, private communications.

# 8 Conclusion

We have proposed a source of coherent Josephson radiation based on a classical pendulum clock. Based on the insight that the limitation on the linewidth of the radiation emitted from the AC Josephson effect originates from the TUR, we utilize the fact that the TUR can be broken by underdamped systems like a pendulum clock. We have designed a superconducting clock circuit that realizes the crucial escapement potential to implement a simple model of a pendulum clock. From a fully quantum mechanical description of the circuit, we derived effective classical equations of motion that showcase a synchronization of the counting and oscillating degrees of freedom. At resonance, the IV curve of the device exhibits a voltage plateau similar to the device presented in [18]. We related this plateau to a violation of the TUR which allows the coherence of the emitted radiation to exceed the limitations of the TUR. The clock circuit is capable of emitting highly coherent AC radiation from a purely DC bias with a coherence and output power that are comparable to other state-of-the-art single-frequency on-chip sources in different material platforms [18]. Furthermore, due to a parallel configuration of the load and the resonator, the properties of the radiation improve with the load impedance which makes the clock circuit a good candidate for on-chip driving in superconducting high-impedance electronics.

Throughout this work, we have focused on the regime, where the non-linear deviations from the ideal escapement coupling can be neglected. A study of those deviations could offer a more complete perspective on the different applications of the clock circuit which might include use as a coherent single photon source. Furthermore, the relation of the TUR to the coherence of Josephson radiation offers a new perspective on on-chip sources which could be an interesting avenue for future research.

# Acknowledgments

We thankfully acknowledge fruitful discussions with S. Lotkhov, F. Kaap and Ç. Ö. Girit.

**Data availability**   The code used to generate the figures is available on Zenodo [38].

**Funding information**   This work was supported by the Deutsche Forschungsgemeinschaft (DFG) under Grant No. HA 7084/6–1.

**Author contributions**   D.S. and F.H. defined the project scope. D.S. developed the code, performed the numerical simulations, and generated the figures. D.S. and J.V. performed the analytical calculations. D.S. wrote the manuscript with input from all the authors. F.H. supervised the project.

# A   Effective description of the circuit

In this appendix, we present the derivation of the effective circuit model used for our simulations. We derive the hybrid description of the clock circuit in terms of a Langevin equation for the counter and a Lindblad equation for the oscillator from a Keldysh path integral description of the circuit presented in Fig. 4. Furthermore, we present the resulting equations of motion in the classical limit which allow to show the suppression of fast rotating terms by the escapement.

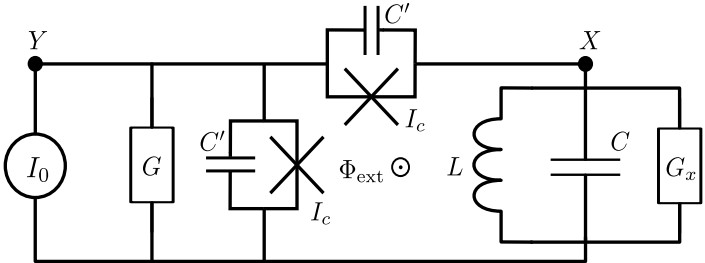

Figure 4: Clock circuit with added junction capacitances $C'$. By including the junction capacitances we can derive a Hamiltonian description of the circuit without the dissipative elements. We can then include the resistive elements in our model by using a Keldysh path integral approach.

## A.1 Circuit Hamiltonian

The circuit has two degrees of freedom given by the flux variables $X$ and $Y$ together with the conjugate charges

$$Q_X = (C + C')\dot{X} - C'\dot{Y}, \qquad Q_Y = 2C'\dot{Y} - C'\dot{X}. \tag{A.1}$$

Note that the junction capacitances are essential for a consistent Hamiltonian description of the circuit without the resistors as they turn $Y$ into a dynamical variable [39]. We introduce the canonical commutators $[\hat{X}, \hat{Q}_X] = [\hat{Y}, \hat{Q}_Y] = i\hbar$ and the transformation

$$\hat{X} = \sqrt{\frac{\hbar Z_0}{2}}(\hat{a}^\dagger + \hat{a}), \qquad \hat{Q}_X = i\sqrt{\frac{\hbar}{2Z_0}}(\hat{a}^\dagger - \hat{a}), \tag{A.2}$$

with the impedance $Z_0 = \sqrt{L/(C + C'/2)}$ and the ladder operators $\hat{a}$ and $\hat{a}^\dagger$ of the harmonic oscillator that fulfill $[\hat{a}, \hat{a}^\dagger] = 1$, as well as a rescaling of the operators $\hat{Y}, \hat{Q}_Y$ via $\hat{y} = 2\pi \hat{Y}/\Phi_0$, $\hat{q} = \hat{Q}_Y/2e$ with $[\hat{y}, \hat{q}] = i$, where $\Phi_0 = 2\pi\hbar/2e$ is the superconducting flux quantum. With the oscillator frequency $\omega_0 = \sqrt{L(C + C'/2)}^{-1}$ and the rescaled external flux $\varphi_{\text{ext}} = 2\pi\Phi_{\text{ext}}/\Phi_0$, we obtain a transformed Hamiltonian given by

$$\hat{H} = \hbar\omega_0\left(\hat{a}^\dagger\hat{a} + \frac{1}{2}\right) + i\frac{\hbar\omega_0}{2}\sqrt{r}(\hat{a}^\dagger - \hat{a})\hat{q} + \frac{\hbar\omega_0}{4}(r + g)\hat{q}^2$$
$$- E_J\left[\cos(\hat{y} - \varphi_{\text{ext}}) + \cos(\hat{y} - \sqrt{r}(\hat{a}^\dagger + \hat{a}))\right] - \frac{\hbar I_0}{2e}\hat{y},$$

with the Josephson energy $E_J = \hbar I_c/2e$ and dimensionless parameters $r = \pi Z_0/R_Q$ and $g = \pi/(\omega_0 R_Q C')$ where $R_Q = 2\pi\hbar/4e^2$ is the quantum resistance. We can obtain a normal ordered Hamiltonian by normal ordering the cosine term using the Baker Campbell Hausdorff formula to obtain

$$\cos(\hat{y} - \sqrt{r}(\hat{a}^\dagger + \hat{a})) = \frac{e^{-\frac{r}{2}}}{2}\left(e^{i\hat{y}}e^{-i\sqrt{r}\hat{a}^\dagger}e^{-i\sqrt{r}\hat{a}} + \text{H.c.}\right). \tag{A.3}$$

With the normal ordered Hamiltonian, we can write down the action for a path integral that combines a coherent state path integral for $X$ with a standard path integral in $y$ and $q$ for $Y$. The resulting action is given by

$$\frac{S}{\hbar} = \int dt \left\{ q\dot{y} + i\bar{\alpha}\dot{\alpha} - \omega_0|\alpha|^2 - \frac{i\omega_0}{2}\sqrt{r}(\bar{\alpha} - \alpha)q - \frac{\omega_0}{4}(r + g)q^2 \right.$$
$$\left. + \frac{I_0}{2e}y + \frac{I_c}{2e}\left[\cos(y - \varphi_{\text{ext}}) + e^{-\frac{r}{2}}\cos(y - \sqrt{r}(\bar{\alpha} + \alpha))\right] \right\}, \tag{A.4}$$

where the second line corresponds to the potential $V(y)$ see below. Note that the characteristic impedance $Z_0$ and frequency $\omega_0$ used in this action go over to the definitions used in the main text in the limit of small junction capacitances $C'/C \ll 1$.

## A.2  Effect of resistive elements

Going over to classical and quantum variables for the $y$ degree of freedom with $y^\pm = y^c \pm y^q/2$, we obtain the Keldysh action where we perform a quadratic expansion in the quantum variables to obtain

$$
\frac{S_K}{\hbar} \approx \int dt \left\{ q^q \left[ \dot{y}^c - \frac{i\omega_0}{4} \sqrt{r} (\bar{\alpha}^+ + \bar{\alpha}^- - \alpha^+ - \alpha^-) - \frac{\omega_0}{2}(r+g)q^c \right] \right.
\tag{A.5}
$$
$$
- y^q \left[ \dot{q}^c - \frac{I_0}{2e} + \frac{I_c}{2e} \left( \sin(y^c - \varphi_{\text{ext}}) + \tfrac{1}{2} e^{-\frac{r}{2}} \left( \sin(y^c - 2\sqrt{r}\,\text{Re}\,\alpha^+) + \sin(y^c - 2\sqrt{r}\,\text{Re}\,\alpha^-) \right) \right) \right]
$$
$$
+ i\bar{\alpha}^+ \dot{\alpha}^+ - i\bar{\alpha}^- \dot{\alpha}^- - \omega_0(|\alpha^+|^2 - |\alpha^-|^2) - i\omega_0\sqrt{r}[\bar{\alpha}^+ - \bar{\alpha}^- - (\alpha^+ - \alpha^-)]q^c
$$
$$
\left. - \frac{I_c}{2e} e^{-\frac{r}{2}} \left[ \cos(y^c - 2\sqrt{r}\,\text{Re}\,\alpha^+) - \cos(y^c - 2\sqrt{r}\,\text{Re}\,\alpha^-) \right] \right\}.
$$

We include two separate environments for the $x$ and $y$ degrees of freedom as independent actions

$$
\frac{S_{G,y}}{\hbar} = -\int dt\,dt' \left[ y^q(t) \frac{Y_y(t-t')}{2\pi G_Q} \dot{y}^c(t') - \frac{i}{2} y^q(t) \frac{K_y(t-t')}{2\pi G_Q} y^q(t') \right],
$$
$$
\frac{S_{G,x}}{\hbar} = -\int dt\,dt' \left[ \frac{1}{2}(\alpha^q(t) + \bar{\alpha}^q(t)) Z_0 Y_x(t-t')(\dot{\alpha}^c(t') + \dot{\bar{\alpha}}^c(t')) \right.
$$
$$
\left. - \frac{i}{4}(\alpha^q(t) + \bar{\alpha}^q(t)) Z_0 K_x(t-t')(\alpha^q(t') + \bar{\alpha}^q(t')) \right],
$$

with the quantum conductance $G_Q = R_Q^{-1}$ and with the time dependent admittance $Y(t)$ and a correlator $K(t) = \int (d\omega/2\pi)\omega \,\text{Re}\, Y_\omega (2n_\omega + 1) e^{-i\omega t}$ where $Y_\omega$ is the Fourier transform of $Y(t)$ and $n_\omega = (\exp(\hbar\omega/k_B T) - 1)^{-1}$ is the Bose-Einstein occupation. We consider Ohmic baths with $Y_{\omega,i} = G_i$ that lead to a time local action for both environments.

## A.3  Equations of motion

In the regime of small damping of the oscillator $Z_0 G_x \ll 1$, we expect the oscillator dynamics to occur at the natural frequency $\omega_0$. We perform a rotating-wave approximation of the action with $\alpha(t) \mapsto \alpha(t)e^{-i\omega_0 t}$ and $\bar{\alpha}(t) \mapsto \bar{\alpha}(t)e^{i\omega_0 t}$, where $\alpha$ and $\bar{\alpha}$ are slow on the scale of $\omega_0$. By first neglecting the obvious fast-rotating terms we simplify the action to

$$
\frac{S_K}{\hbar} \approx \int dt \left\{ q^q \left[ \dot{y}^c - \frac{\omega_0}{2}(r+g)q^c \right] - y^q \left[ \dot{q}^c - \frac{I_0}{2e} \right. \right.
\tag{A.6}
$$
$$
\left. + \frac{I_c}{2e} \left( \sin(y^c - \varphi_{\text{ext}}) + \tfrac{1}{2} e^{-\frac{r}{2}} \left( \sin(y^c - 2\sqrt{r}\,\text{Re}(\alpha^+ e^{-i\omega_0 t})) + \sin(y^c - 2\sqrt{r}\,\text{Re}(\alpha^- e^{-i\omega_0 t})) \right) \right) \right]
$$
$$
\left. + i\bar{\alpha}^+ \dot{\alpha}^+ - i\bar{\alpha}^- \dot{\alpha}^- - \frac{I_c}{2e} e^{-\frac{r}{2}} \left( \cos(y^c - 2\sqrt{r}\,\text{Re}(\alpha^+ e^{-i\omega_0 t})) - \cos(y^c - 2\sqrt{r}\,\text{Re}(\alpha^- e^{-i\omega_0 t})) \right) \right\},
$$

where we also expanded in the quantum variable of the counter to quadratic order. This is justified in the regime of large damping $G_y = G \gg G_Q$ of the counter, where the potential,

$$
V(y) = \frac{\hbar}{2e} \left\{ I_0 y + I_c \left[ \cos(y - \varphi_{\text{ext}}) + e^{-\frac{r}{2}} \cos(y - 2\sqrt{r}\,\text{Re}\,\alpha) \right] \right\},
\tag{A.7}
$$

fulfils $|V'''(y)| \ll G|V'(y)|/(2\pi G_Q)$ [12, 37, 40]. For strong damping in the $y$-degree of freedom, we expect $y^c$ to be dominated by a linear growth in time. We make the ansatz

$$y^c = \omega_0 t + \theta \,, \tag{A.8}$$

with a slow phase variable $\theta$ that fulfills $\dot\theta \ll \omega_0$. Note that we neglected any fast rotating contribution to $y^c$, which, as we show later on, is valid at an external magnetic flux of half a flux quantum in the regime of small $r$. Due to this, $\sin(y^c - \varphi_{\text{ext}})$ only gives a fast-rotating contribution to the action and can be neglected. In addition, we introduce a shifted classical momentum

$$q^c = \frac{2}{r+g} + p^c \,. \tag{A.9}$$

Using a Jacobi-Anger expansion, this ansatz allows for the clear identification of the remaining fast-rotating terms resulting in

$$\sin(y^c - 2\sqrt{r}\,\text{Re}(\alpha e^{-i\omega_0 t})) \approx -J_1(2\sqrt{r}|\alpha|)\cos(\theta - \varphi)\,,$$
$$\cos(y^c - 2\sqrt{r}\,\text{Re}(\alpha e^{-i\omega_0 t})) \approx J_1(2\sqrt{r}|\alpha|)\sin(\theta - \varphi)\,,$$

with the Bessel function $J_1$ of the first kind and $\alpha = |\alpha|e^{-i\varphi}$. Note that this only requires the phase difference $\theta - \varphi$ to be a slow variable and not the individual phases. The resulting action is given by

$$\frac{S_K}{\hbar} \approx \int dt \left\{ q^q \left[ \dot\theta - \frac{\omega_0}{2}(r+g)p^c \right] - y^q \left[ \dot p^c - \frac{I_0}{2e} + \frac{1}{\hbar}\frac{\partial}{\partial\theta}\left(\mathcal{H}_{\text{RW}}(\alpha^+, \bar\alpha^+, \theta) + \mathcal{H}_{\text{RW}}(\alpha^-, \bar\alpha^-, \theta)\right) \right] \right.$$
$$\left. + i\bar\alpha^+\dot\alpha^+ - i\bar\alpha^-\dot\alpha^- - \mathcal{H}_{\text{RW}}(\alpha^+, \bar\alpha^+, \theta) + \mathcal{H}_{\text{RW}}(\alpha^-, \bar\alpha^-, \theta) \right\}\,. \tag{A.10}$$

With a rotating-wave Hamiltonian function $\mathcal{H}_{\text{RW}}(\alpha, \bar\alpha, \theta)$

$$\frac{\mathcal{H}_{\text{RW}}(\alpha, \bar\alpha, \theta)}{\hbar} = i\frac{I_c}{2e}e^{-\frac{r}{2}}\frac{J_1(2\sqrt{r}|\alpha|)}{2|\alpha|}\left(\alpha e^{i\theta} - \bar\alpha e^{-i\theta}\right)\,,$$

from which we can obtain the rotating-wave Hamiltonian by reinserting the ladder operators. We obtain

$$\frac{\hat{H}_{\text{RW}}(\theta)}{\hbar} = i\frac{I_c}{4e}e^{-\frac{r}{2}}:\frac{J_1(2\sqrt{r\hat{a}^\dagger\hat{a}})}{\sqrt{\hat{a}^\dagger\hat{a}}}(\hat{a}e^{i\theta} - \hat{a}^\dagger e^{-i\theta}):\,, \tag{A.11}$$

where $:\cdot:$ denotes the normal ordering of the ladder operators. With this form and the environmental action of the oscillator

$$\frac{S_{G,x}}{\hbar} \approx i\gamma \int dt \left[ -n_0\bar\alpha^+\alpha^- - (n_0 + 1)\bar\alpha^-\alpha^+ + (n_0 + \tfrac{1}{2})(|\alpha^+|^2 + |\alpha^-|^2) \right]\,,$$

the reduced density matrix $\rho$ of the harmonic oscillator follows the Lindblad equation [41,42] given in Eq. (5) with a small damping constant $\gamma = \omega_0 Z_0 G_x \ll \omega_0$ and the Bose-Einstein occupation $n_0$ at frequency $\omega_0$. We rewrite the environmental action of the $y$-degree of freedom as

$$\frac{S_{G,y}}{\hbar} = \int dt\, y^q(t)\left[ \xi(t) - \frac{G}{2\pi G_Q}(\omega_0 + \dot\theta(t)) \right]\,, \tag{A.12}$$

using a Hubbard-Stratonovich transform to introduce the Gaussian random variable $\xi$ with $\langle\xi(t)\rangle = 0$ and $\langle\xi(t)\xi(t')\rangle = k_B T G\delta(t - t')/(\pi\hbar G_Q)$. Integrating over the quantum variables for $y$ yields a Langevin equation [40]

$$\frac{2}{\omega_0(r+g)}\ddot\theta + \frac{G}{2\pi G_Q}(\omega_0 + \dot\theta) = \frac{I_0}{2e} - \frac{1}{\hbar}\left\langle \frac{\partial\hat{H}_{\text{RW}}(\theta)}{\partial\theta} \right\rangle_\rho + \xi(t)\,, \tag{A.13}$$

where $\langle \hat{O}\rangle_\rho = \text{tr}[\hat{O}\hat{\rho}]$ denotes the expectation value with respect to the density matrix of the harmonic oscillator that is still conditioned on the random force $\xi$. When solving the coupled Lindblad and Langevin equation, we integrate both equations for a given noise trajectory by updating $\hat{\rho}$ and $\theta$ in each time step and then average over the noise $\xi$ in the end. In the path integral setting, this corresponds to first integrating out the quantum variables for $y$ and then integrating both the classical variables for $y$ and the $x$-variables in each time step. The integral over the noise $\xi$ is performed last. In the limit of small Junction capacitances $g \gg 1$, the equation becomes overdamped and we obtain Eq. (6) by a multiplication of both sides with $2\omega_0 r$.

## A.4 Adler-type equations in the classical limit

In the regime of large photon numbers and small light-matter coupling, we can simplify the Bessel function to obtain the linear Hamiltonian given in Eq. (7). This linear form allows us to formulate a closed set of equations for the expectation values of the ladder operators of the oscillator. For the annihilation operators we obtain

$$\frac{d}{dt}\langle \hat{a}\rangle_\rho = -\frac{\gamma}{2}\langle \hat{a}\rangle_\rho - \frac{\kappa}{4\sqrt{r}\omega_0}e^{-i\theta}, \qquad \frac{d}{dt}\langle \hat{a}^2\rangle_\rho = -\gamma\langle \hat{a}^2\rangle_\rho - \frac{\kappa}{2\sqrt{r}\omega_0}\langle \hat{a}\rangle_\rho e^{-i\theta},$$

with the conjugate equations for the creation operators and an equation

$$\frac{d}{dt}\langle \hat{n}\rangle_\rho = -\gamma\langle \hat{n}\rangle_\rho + \gamma n_0 - \frac{\kappa}{4\sqrt{r}\omega_0}(\langle \hat{a}\rangle_\rho e^{i\theta} + \langle \hat{a}^\dagger\rangle_\rho e^{-i\theta}), \tag{A.14}$$

for the photon number. In the steady state, we obtain

$$\langle \hat{a}^2\rangle_\rho = \frac{\kappa^2}{4r\omega_0^2\gamma^2}e^{2i\theta}, \tag{A.15}$$

which is proportional to the maximum coherent photon number in the steady state given by Eq. (8). Therefore $r\langle \hat{a}^2\rangle_\rho \ll 1$ holds provided that the circuit is in the regime of weak light matter coupling $rn_{\text{coh}} = \kappa^2/(4\omega_0^2\gamma^2) \ll 1$. In order to analyze the dynamics of the circuit we make the ansatz $2\sqrt{r}\langle \hat{a}\rangle_\rho = 2\sqrt{r}\langle \hat{a}^\dagger\rangle_\rho^* = A(t)e^{-i\varphi(t)}$ with slow real variables $A$ and $\varphi$ to obtain the classical set of Adler-type equations given in Eq. (9).

## A.5 Suppression of fast rotating terms

In addition to the Langevin equation at zero frequency, we obtain an equation at frequency $\omega_0$, which may lead to a fast rotating contribution of the counter $y^c$ which we denote by $\theta_f$. In first order perturbation theory, we obtain

$$\dot{\theta}_f = \frac{\kappa}{\Gamma}\Bigg[\sin(\omega_0 t + \theta - \varphi_{\text{ext}}) + e^{-\frac{r}{2}}\Big\langle :J_0(2\sqrt{r\hat{a}^\dagger\hat{a}}): \Big\rangle_\rho \sin(\omega_0 t + \theta)$$
$$+ ie^{-\frac{r}{2}}\Big\langle :\frac{J_2(2\sqrt{r\hat{a}^\dagger\hat{a}})}{\hat{a}^\dagger\hat{a}}(\hat{a}^2 e^{-i(\omega_0 t - \theta)} - \text{H.c.}): \Big\rangle_\rho\Bigg],$$

where we neglected the noise terms at frequency $\pm\omega_0$, since they have no contribution on average. We make the ansatz $\theta_f = -\text{Im}[Be^{-i(\omega_0 t + \psi)}]$ with slow and real amplitude $B$ and phase $\psi$. In the regime of small light matter coupling $\langle r\hat{a}^\dagger\hat{a}\rangle_\rho \ll 1$ we expand the Bessel functions up to linear order in $r$ to obtain

$$Be^{-i\psi} = \frac{\kappa}{\Gamma\omega_0}\Big[(1 + e^{i\varphi_{\text{ext}}})e^{-i\theta} + 2ir\langle \hat{a}^2\rangle_\rho e^{i\theta}\Big]. \tag{A.16}$$

At $\phi_{\text{ext}} = \pm\pi/2$, we can suppress the leading contribution to the oscillation to realize an escapement coupling. This corresponds to a DC-SQUID with maximum frustration which exhibits no Josephson oscillations. The first non-zero correction is suppressed provided that $r\langle \hat{a}^2 \rangle_\rho$ is small, which is fulfilled for the classical regime. Therefore, in the classical regime, the counter exhibits linear growth in time without a substantial oscillating behavior.

## B  Classical equations in the small impedance regime

In this appendix, we show how the Adler-type equations without noise arise from the classical model of a pendulum clock. Furthermore, we include the thermal noise in the synchronization regime to calculate the variances of the counter and the oscillator phase.

### B.1  Adler-type equations from the classical model

In order to show the equivalence between our minimal model of a classical pendulum clock and the clock circuit, we show that the classical equations of motion

$$\Gamma\dot{Y} = f + \kappa X\cos Y, \qquad \ddot{X} + \gamma\dot{X} + \omega_0^2 X = \kappa\sin Y, \tag{B.1}$$

can be used to derive the effective Adler-type equations for the phase dynamics up to the thermal noise term $\xi$. We insert the ansatz of a regulated linear motion $Y = \omega_0 t + \theta(t)$ for the counter and an oscillation with a slow time-dependent amplitude and phase $\dot{A}, \dot{\varphi} \ll \omega_0$ for the oscillator $X = \text{Re}[A(t)e^{-i(\omega_0 t + \varphi(t))}]$. From this, we perform a rotating-wave approximation, where we neglect the fast-rotating terms in the equation of the counter. In the equation for the oscillator, we obtain many terms that include the small quantities $\dot{A}$, $\dot{\varphi}$ and $\gamma$. We only consider the dominant contributions that also include a factor $\omega_0$ to obtain

$$\dot{\theta} = \Delta\omega + \frac{\kappa}{2\Gamma}A\cos(\varphi - \theta), \qquad \dot{A} + i\dot{\varphi}A = -\frac{\gamma}{2}A - \frac{\kappa}{2\omega_0}e^{-i(\varphi - \theta)}, \tag{B.2}$$

as for the clock circuit without thermal noise.

### B.2  Phase synchronization with thermal noise

In the case of synchronization, the phase difference $\varphi - \theta$ becomes independent of time, which allows us to neglect the contribution from $\dot{A}$ to obtain the steady state solution without noise presented in Eq. (10). To account for the effects of the thermal fluctuations, we consider small perturbations $\delta\varphi$, $\delta\theta$ and $\delta A$ which fulfill

$$\delta\dot{\theta} = \frac{\kappa}{2\Gamma}[\cos(\varphi_0 - \theta_0)\delta A - A_0\sin(\varphi_0 - \theta_0)(\delta\varphi - \delta\theta)] + \xi,$$
$$\delta\dot{\varphi}A_0 + \dot{\varphi}_0\delta A = \frac{\kappa}{2\omega_0}\cos(\varphi_0 - \theta_0)(\delta\varphi - \delta\theta),$$
$$\delta\dot{A} = -\frac{\gamma}{2}\delta A + \frac{\kappa}{2\omega_0}\sin(\varphi_0 - \theta_0)(\delta\varphi - \delta\theta).$$

In the regime of small light-matter coupling, the perturbations of the Amplitude relax fast and can be neglected. We therefore consider the regime of constant amplitude $A = A_0$, in which the reduced equations are given by

$$\begin{pmatrix} \delta\dot{\theta} \\ \delta\dot{\varphi} \end{pmatrix} = \frac{\gamma}{2}\begin{pmatrix} -\beta & \beta \\ 1 & -1 \end{pmatrix}\begin{pmatrix} \delta\theta \\ \delta\varphi \end{pmatrix} + \begin{pmatrix} \xi(t) \\ 0 \end{pmatrix}. \tag{B.3}$$

The homogeneous part of the equation is solved by

$$\begin{pmatrix} \delta\theta \\ \delta\varphi \end{pmatrix} = c_1 \begin{pmatrix} 1 \\ 1 \end{pmatrix} + c_2 e^{-\frac{\gamma}{2}(1+\beta)t} \begin{pmatrix} -\beta \\ 1 \end{pmatrix}, \tag{B.4}$$

where $c_1$ and $c_2$ are integration constants. By solving the inhomogeneous equation via variation of constants, we determine the variance of the light phase $\langle\langle\delta\varphi^2\rangle\rangle$ and the variance of the counter $\langle\langle\delta\theta^2\rangle\rangle$. We obtain

$$\langle\langle\delta\varphi^2\rangle\rangle = \langle c_1^2 \rangle + 2\langle c_1 c_2\rangle e^{-\frac{\gamma}{2}(1+\beta)t} + \langle c_2^2\rangle e^{-\gamma(1+\beta)t},$$

$$\langle\langle\delta\theta^2\rangle\rangle = \langle c_1^2 \rangle - 2\beta\langle c_1 c_2\rangle e^{-\frac{\gamma}{2}(1+\beta)t} + \beta^2\langle c_2^2\rangle e^{-\gamma(1+\beta)t},$$

where parameters $c_1$ and $c_2$ are defined by

$$c_1 = \int \frac{\xi(t)}{1+\beta} dt, \quad \text{and} \quad c_2 = -\int \frac{\xi(t)}{1+\beta} e^{\frac{\gamma}{2}(1+\beta)t} dt.$$

We calculate the expectation values using the correlator $\langle\xi(t)\xi(t')\rangle = \epsilon\delta(t-t')$ and consider only the terms that increase with time which yields

$$\langle c_1^2 \rangle = \frac{\epsilon t}{(1+\beta)^2}, \qquad \langle c_2^2 \rangle = \frac{\epsilon}{\gamma(1+\beta)^3}\left(e^{\gamma(1+\beta)t}-1\right), \qquad \langle c_1 c_2\rangle = -\frac{2\epsilon}{\gamma(1+\beta)^3}\left(e^{\frac{\gamma}{2}(1+\beta)t}-1\right).$$

The variances $\langle\langle\varphi^2\rangle\rangle$ and $\langle\langle\theta^2\rangle\rangle$ are therefore given by

$$\begin{aligned}
\langle\langle\varphi^2\rangle\rangle &= \frac{\epsilon}{\gamma(1+\beta)^3}\left\{\gamma(1+\beta)t - \left[3 + e^{-\gamma(1+\beta)t} - 4e^{-\frac{\gamma}{2}(1+\beta)t}\right]\right\}, \\
\langle\langle\theta^2\rangle\rangle &= \frac{\epsilon}{\gamma(1+\beta)^3}\left\{\gamma(1+\beta)t + \beta\left[4 - 4e^{-\frac{\gamma}{2}(1+\beta)t} + \beta - \beta e^{-\gamma(1+\beta)t}\right]\right\}.
\end{aligned} \tag{B.5}$$

In the limit of long times $\gamma(1+\beta)t \gg 1$, the linearly growing terms yield the estimates for the TUR and the quality factor of the radiation given in the main text.

## C  Asymmetric critical current

The escapement coupling, we implemented with our circuit, relies on the fact that the zeroth order contributions in $X$ cancel at an external flux of half a flux quantum for the potential shown in Eq. (4). This full cancellation however can only occur if both junctions have the same critical current $I_c$. In case of an experimental implementation this exact match in critical currents cannot be guaranteed. In order to analyze the effect of asymmetric critical currents, we consider the classical equations with a modified escapement potential

$$V_c = -\kappa[(1+\delta)\cos(Y-X) - \cos(Y)], \tag{C.1}$$

where the critical current of the junction connected to the oscillator deviates from $I_c$ by a factor of $(1+\delta)$. In Fig. 5, we show the numerical IV-curves resulting from the classical equations of motion with the modified escapement potential with the same circuit parameters as for Fig. 3. Since the lowest order contribution does not cancel anymore, the curves with asymmetric critical current exhibit an additional supercurrent region similar to the one of a single Josephson junction. This added supercurrent region competes with the synchronization plateau, which decreases the size of the synchronization region while also increasing the corresponding differential resistance. Since we expect a direct correspondence between the differential resistance on the synchronization plateau and the breaking of the TUR, we expect that the coherence of the resulting radiation will also be reduced. The curves show that up to a deviation of $\delta = \pm 0.1$, the circuit still exhibits a clock resonance with a flat plateau that likely also corresponds to highly coherent radiation.

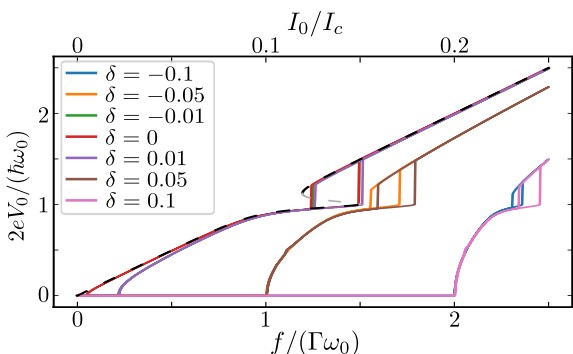

Figure 5: IV-curves with an asymmetric critical current, where the critical current of the junction connected to the oscillator is given by $I_c' = (1 + \delta)I_c$. The other circuit parameters are the same as the ones used in Fig. 3. While there is an additional supercurrent region arising that competes with the plateau of clock resonance. The clock resonance is still present with a steeper plateau even up to $\delta = 0.1$. From this, we conclude that the circuit will likely still emit radiation with a coherence that is increased above the limit set by the TUR.

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
