# Peer review of "The superconducting clock-circuit: Improving the coherence of Josephson radiation beyond the thermodynamic uncertainty relation"

_SciPost Physics, doi:SciPost Phys. 17, 140 (2024)_

## Round 1 · Referee Report · Anonymous (Referee 1) · 2024-8-17

Strengths
1—Given a direct connection between a well-known limitation in the coherence of the AC Josephson effect and the Thermodynamic Uncertainty Relation (TUR).
2— Provided a minimal setup for the realization of a superconducting circuit able to produce coherent radiation beyond standard TUR limits.
3—The introduction is well-written and clearly defines the context and goal of the paper.
Weaknesses
1- The current presentation in Section 4 (and partially in Section 5) may be a little obscure due to some details only present in the Appendices - see also the report for some suggestions.
Report
The authors theoretically investigate a superconducting circuit for the on-chip generation of coherent Josephson radiation from a purely DC input bias. The AC Josephson effect naturally converts a DC voltage bias into an oscillating signal, but the coherence of such radiation is limited by the Thermodynamic Uncertainty Relation (TUR). Inspired by recent work on TUR violation in a classical pendulum clock [Ref. 30], the authors propose a superconducting clock circuit in which the violation of TUR determines a lower bound on the linewidth of the emitted radiation.
More precisely, the authors consider a minimal model for a pendulum clock and investigate a circuit with two degrees of freedom. In their setup, a Josephson junction with a real voltage bias (counter) is coupled to an underdamped resonator (oscillator) via an identical Josephson junction. Using an external magnetic field in the loop, such a configuration at half-flux quantum mimics the escapement potential of the pendulum clock necessary to overcome the TUR limitation in the overdamped dynamics of the counter.
On the technical side, the authors solve the quantum dynamics of the counter-oscillator system using a Lindblad equation for the oscillator and a Langevin approach for the counter. The details of the model are discussed in Appendix A, and some analytical approximations and classical limits are provided in the main text. The authors identify a synchronization strength parameter $\beta$, which provides a figure of merit for the generation of coherent radiation beyond the TUR limit for a realistic voltage bias Josephson junction.
This work can stimulate further theoretical research on this topic and possibly be relevant for the experimental implementation of on-chip coherent radiation in high-impedance environments. As such, I would recommend this article for publication in SciPost after addressing the comments below.
Requested changes
1— In section 3, the escapement potential follows from Eq.(4) at half flux quantum in the limit of small oscillation amplitude. The cancellation of the $\cos(Y)$ term at the leading order expansion in $X\ll 2\pi$ requires the two junctions to be identical. How robust is the setup to the asymmetry between the two junctions possibly arising from fabrication? Does the asymmetry generally tighten the bounds on the quality factor of the radiation? It would be good to mention these points in the discussion.
2—To improve readability for a broader audience and make the article more self-contained, I would consider expanding Sec. II concerning the classical model of a pendulum clock. This choice could give more insight, for instance, of the ansatz $Y(t)=\omega_0 t+\theta(t)$ made in Sec. IV for the quantum regime. Moreover, the direct derivation of Adler-type equations (9) from Eq.(3) could be provided in some Appendix or in alternative some relevant citation could be added on this point. On another note, I would add some details on the continuous model for the escapement potential and maybe provide (even only as a footnote or in the appendix) the explicit connection of the potential $V_C(X, Y)$ and the contents of Ref. [30].
3—Section 4 (and similarly Section 5) is somewhat harder to follow compared to the rest of the main text since some of the results heavily depend on the derivation given in the Appendix. One solution would be to break down Appendices A and B in some subsections and add more references to them when quoting some of the relevant results of Section 4. Moreover, I would make more explicit which set of equations is solved when doing ''simulation ... for the full quantum mechanical model" compared to analytics. On a final note: is there some intuition on the increasing size of the voltage plateaux by increasing the light-matter coupling $r$?
4— Some technical comments: does the dephasing rate take the form of Eq. (1) only under the assumption $k_B T\gg eV$? If so, it would be good to specify it. After Eq.(23), should it be $\dot\theta\ll \omega_0$ rather than $\dot\theta\lesssim \omega_0$? Finally, I would provide some references (even to relevant books) for the path-integral discussion and other technical points arising after Eq.(18) in Appendix A. This could help the non-expert reader.
5— For completeness, add explicitly the definition of some symbols and
notation in the text:
-define the electric charge $e$ and Planck's constant $h$ (and/or the
reduced one $\hbar$) in the Introduction when these symbols are first
introduced in the paper;
-define the notation on the brackets: $\langle\langle Y^2\rangle\rangle$
for the variance and $\langle Y \rangle$ for the expectation value (here
both concerning time averaging, I think);
-define $\kappa$ as the coupling parameter in section 2.
-add a reference to Appendix A when discussing the small junction
capacitance limit before Eq.(6) since they appear only in the schematic of
Fig.4 and not in Fig.1 of the main text
-The definition of $Z_0$ slightly differs in the main text and in Appendix A
after Eq. (17), where the latter also contains the junction capacitance.
6— In Fig.2 and Fig.3, I would consider using a double axis for the x- coordinate adding the corresponding value of the current (in units of $I_c$ or $I_0$). What is the value of $\beta$ for the dotted curves in Figs. 3b and 3c? Morevoer I would add "(not shown)" after "Off resonance, $Q$ is of the order
of $10^{-9}$..." in the caption of Fig.3.
7—Few typos:
-a factor $4\pi$ is likely missing in the denominator of the quality factor
$Q_0$ - see paragraph following Eq. (1) in the Introduction;
-in the text above Eq. (18), I think one should replace $Z\rightarrow Z_0$ in
the definition of the coupling parameter $r$;
-"light-mater" $\rightarrow$"light-matter" in the text before Eq. (29);
-Again, in the text before Eq. (29), a sentence seems broken: "for the
expectation values of.";
-after Eq.(30), there is likely a typo in $r n_{coh}=
\kappa/(4\omega_0\gamma)$, where $\omega_0$ and $\gamma$
should be squared according to Eq. (8).
Recommendation
Ask for minor revision
Author: David Scheer on 2024-09-23 [id 4798]
(in reply to Report 1 on 2024-08-17)
Errors in user-supplied markup (flagged; corrections coming soon)
We thank the referee for their thorough reading of the manuscript and the detailed suggestions for improvement of the manuscript.
The referee outlined many possible improvements to the manuscript which we want to address individually:
* * *
1— In section 3, the escapement potential follows from Eq.(4) at half flux quantum in the limit of small oscillation amplitude. The cancellation of the
$\cos(Y)$ term at the leading order expansion in $X\ll 2\pi$ requires the two junctions to be identical. How robust is the setup to the asymmetry between the two junctions possibly arising from fabrication? Does the asymmetry generally tighten the bounds on the quality factor of the radiation? It would be good to mention these points in the discussion.
While the best results for coherence in the clock resonance are reached for symmetrical critical currents, the resonance itself is robust to asymmetrical junctions. Up to an asymmetry of about 10%, the voltage plateau
is still present in the IV curve even if it is reduced by a competition with the supercurrent region added by the
asymmetry.
We included a short discussion of the asymmetry to Sec. VII and added an Appendix with classical simulation results
including asymmetric junctions.
* * *
2—To improve readability for a broader audience and make the article more self-contained, I would consider expanding Sec. II concerning the classical model of a pendulum clock. This choice could give more insight, for instance, of the ansatz $Y(t)=\omega_0 t+\theta (t)$ made in Sec. IV for the quantum regime. Moreover, the direct derivation of Adler-type equations (9) from Eq.(3) could be provided in some Appendix or in alternative some relevant citation could be added on this point. On another note, I would add some details on the continuous model for the escapement potential and maybe provide (even only as a footnote or in the appendix) the explicit connection of the potential $V_C(X,Y)$ and the contents of Ref. [30].
We agree that a more extensive discussion of the classical model will increase the readability as it
helps to build an intuition for the dynamics of the circuit.
We added a discussion of the ansatz for the dynamics of the counter and the oscillator to Sec. II
which we think also helps to understand the nature of the resonant motion in the clock. We included a
derivation of the resulting Adler-type equations as a part of Appendix B. With regards to the escapement
potential, we added a footnote in Sec. II that discusses in detail the relation of our escapement to the one
presented by Pietzonka.
* * *
3—Section 4 (and similarly Section 5) is somewhat harder to follow compared to the rest of the main text since some of the results heavily depend on the derivation given in the Appendix. One solution would be to break down Appendices A and B in some subsections and add more references to them when quoting some of the relevant results of Section 4. Moreover, I would make more explicit which set of equations is solved when doing ''simulation ... for the full quantum mechanical model" compared to analytics. On a final note: is there some intuition on the increasing size of the voltage plateaux by increasing the light-matter coupling $r$?
We agree that the two sections benefit from a closer linkage to the appendices as this makes the results
more tractable and allows for a more targeted reading of the appendix. We attribute the increasing size of the voltage
plateaus to the overall increase of the light-matter interaction strength, which facilitates synchronization effects.
We restructured the Appendices A and B so that the relevant derivation of the effective circuit model is now fully contained in Appendix A. As the referee suggested, we introduced smaller sections especially in Appendix A which we refer to when used for the main text. We think that this increased the overall readability of sections 4 and 5 as well as of the appendices themselves. We added a comment of the effect of the light-matter coupling on the width of the voltage plateaus at the end of section 4. Furthermore, we replaced the vague referral to the “full quantum mechanical model” by a specification that we simulate the coupled Lindblad and Langevin equations using the full rotating-wave Hamiltonian presented in Appendix A3.
* * *
4— Some technical comments: does the dephasing rate take the form of Eq. (1) only under the assumption
$k_BT\gg eV$? If so, it would be good to specify it. After Eq.(23), should it be $\dot\theta\ll\omega_0$
rather than $\dot\theta\lesssim\omega_0$?
Finally, I would provide some references (even to relevant books) for the path-integral discussion
and other technical points arising after Eq.(18) in Appendix A. This could help the non-expert reader.
We thank the referee for pointing this out as this assumption appeared necessary based on the cited material.
While the derivation in the paper by Dahm et al. (1969), which we cite
for the dephasing rate, relies on the limit of large temperatures for the down-conversion of high frequency
noise, the dephasing rate of the Josephson oscillations can also be understood in terms of the DC contribution
of the Johnson-Nyquist noise. For this DC contribution, the characteristic frequency is zero. Therefore, it is always
in the large temperature limit. For a more complete discussion in the RSJ-model, also see the book by Likharev
“Dynamics of Josephson junctions and circuits” (1986) which we also cite in the updated version of the paper.
After Eq.(23), the limit $\dot\theta\ll\omega_0$ is more appropriate as it is relevant for the clock-resonance.
While some analytical results also show good agreement to the numerics outside of this limit, they are still all
derived under this assumption. We adapted this in the new version.
We added further references on path integrals with a focus on the relation to the Langevin and Lindblad equation
to the discussion in Appendix A.
* * *
5-7- The referee brought to attention typos in the manuscript, some incomplete information
for Figures 2. and 3. as well as missing definitions of relevant symbols in the text.
We fixed all of the addressed points in the new version of the manuscript. Most importantly,
we specified the definitions of the different averaging processes with a clear distinction
between averaging over realizations of the thermal noise (e.g. $\langle Y\rangle$) and expectation
values of operators with respect to the density matrix of the oscillator.
Author: David Scheer on 2024-09-23 [id 4799]
(in reply to Report 2 on 2024-08-29)

---

## Round 1 · Referee Report · Anonymous (Referee 2) · 2024-8-29

Strengths
1- links stochastic thermodynamics (specifically TURs and their breakdown) to the mesoscopic physics of superconducting devices
2-describes the physics of a device that is amenable to practical implementation and can have an impact on current quantum technology
Weaknesses
1- some important details are relegated to appendices or missing
Report
I think the paper meets the expectation criterium of "providing a novel and synergetic link between different research areas": stochastic thermodynamic and mesoscopic physics of superconducting devices.
After revision I expect the paper to meet all the general acceptance criteria. See requested changes below
Requested changes
1- While the paper is clear and well written, it can improve further in regard to better integrate the content of the appendix in the main text. Maybe the appendices can be inserted in the main text? (optional)
2- How is the entropy production rate sigma calculated? Since the main message of the ms is that the device beats the TUR bound, which is expressed in terms of entropy production rate, it is crucial that the estimation of the entropy production rate be detailed. I could not find that in the text/appendices
Recommendation
Ask for minor revision

---

## Round 2 · Referee Report · Anonymous (Referee 1) · 2024-10-7

Report

In the revised manuscript, the authors have addressed all my comments. I recommend this manuscript for publication in SciPost Physics.

Recommendation

Publish (meets expectations and criteria for this Journal)

---

## Round 2 · Author Response

We again thank the referees for their comments. We implemented the requests for changes of the manuscript as well as a few additional small improvements. A comprehensive list of changes is given below.

---

## Round 2 · List of Changes

-Added clarification of different expectation values where $\langle.\rangle$ denotes averages with respect to thermal fluctuations and $\langle.\rangle_\rho$ denotes the expectation value with regards to the density matrix of the oscillator that is still conditioned on the thermal fluctuations. -Added further comments to clarify the entropy production rate as well as the form of the escapement potential with regards to Ref. [25]. -Included a paragraph on the ansatz for the classical model to build an intuition for the clock dynamics -Included Ref. [36,37] as a study of a similar circuit. -Restructured the Appendices into smaller subsections to allow for a more clear presentation in sections 4 and 5 with references to individual sections. Appendix A now shows the complete derivation of the effective quantum mechanical model of the circuit and Appendix B covers the Adler-type equations in the classical model as well as the influence of thermal noise on the synchronization. -Extended the introductory paragraph of section 4 to briefly outline the use of a Keldysh path integral description -Specified the simulated model as the coupled Lindblad and Langevin equations using the full rotating wave Hamiltonian from Appendix A. -Added interpretation for the increasing size of the synchronization plateau with increasing light-matter coupling $r$. -Added second x-axis to the Figures 2. and 3. to explicitly show the dependence on the bias current. -Included discussion of asymmetric critical currents in section 7. -Added references to path integral literature in Appendix A. -Added Appendix C to further discuss the case of asymmetric critical currents. - Added missing definitions of constants -Corrected typos

---

## Editorial Decision

published